# Leveraging Ontologies for Knowledge Graph Schemas

Daniela Oliveira, Ratnesh Sahay, and Mathieu d'Aquin

Insight Centre for Data Analytics, Data Science Institute, NUI Galway, Ireland
`firstname.lastname@insight-centre.org`

**Abstract.** Knowledge Graphs have emerged as a core technology to enhance the search experience of users in research and industry alike. Applications are being developed to build and exploit graph structures in data repositories, to improve search and integration of knowledge. However, data is available on the Web in diverse formats and integrating data from different sources is still an open research area. In this paper, we propose a method to build a schema for Knowledge Graphs that leverages ontologies by integrating them in a single unified graph, enriched by an ontology matching step. We evaluate this method by comparing the different stages of graph structures and measuring structural properties of the resulting graphs. We also propose an approach that groups ontology classes by conceptual type, showcasing how an underlying ontology-based schema can enhance a Knowledge Graph. The results demonstrate the viability of this approach, highlighting how the chosen techniques based on edge addition successfully make our ontology-based knowledge graph schema more complete and tightly connected.

**Keywords:** Knowledge Graphs · Semantic Web · Ontologies

## 1  Introduction

The Web provides an infrastructure where knowledge can be stored, shared, and created, commonly in an unstructured format, using natural language. In response to this lack of structure, the Semantic Web [2] and Linked Open Data (LOD) [3] emerged to structure and integrate data on the Web, with standardised formats that can be read and interpreted by both machines and humans alike.

Search engines often find unstructured information just by crawling the Web. Recently, however, search engines have been evolving towards a more developed search experience by leveraging Semantic Web technologies to structure information. For instance, Google coined the term *Knowledge Graph*[1] (KG) and described it as a way to find new information about entities quickly and easily. The concept of KG has since been adopted by the Semantic Web community to describe graphs that aggregate knowledge from different domains by including real-world entities and their relationships [18] (e.g. DBPedia [12] and YAGO [20]).

However, creating KGs from information on the Web, even when structured, is not a trivial task. When information sources were not specifically developed with interoperability in mind, data structures do not follow the same standards

---

[1] `https://googleblog.blogspot.com/2012/05/introducing-knowledge-graph-things-not.html` (Retrieved in November 2018)

or models, and integration becomes a tough challenge. Therefore, ontologies are fundamental to ensure the consistency between datasets since they formally represent and precisely define concepts and their relationships. When combined with LOD, they improve the semantic content of the data and link datasets at a schema level (as shown, for example, in [5] for the education domain).

Figure 1 illustrates a well-connected network of ontology classes that is used at the schema level to integrate knowledge in a KG. Ontologies are structured as a graph where the concepts/classes are nodes and the axioms/properties are edges. These axioms refer to relationships within the ontology (intra-links) or link to concepts in other ontologies (inter-links). Then, graph enrichment methods facilitate the discovery of new relationships between ontology entities (enrichment links), resulting in a more tightly connected graph. Finally, assigning classes from the schema layer to entities in the data layer (annotation links) allows the inference of new relationships or paths between entities (inferable links).

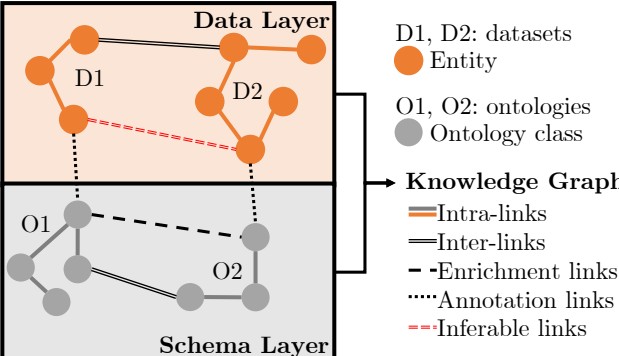

Fig. 1: KG with two layers illustrating different links.

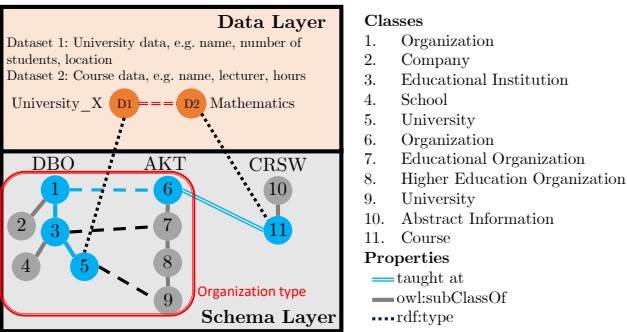

Fig. 2: Example of use of the ontology graph schema to structure data in a KG.

Figure 2 presents an hypothetical example of how multiple ontologies can be connected in a single graph and exploited as a KG schema. The Schema Layer contains a graph with three example ontologies: DBPedia, AKT[2], and CRSW[3]. Now lets consider the task of two datasets that need to be integrated into the KG: one describing universities and the other describing courses. After align-

---

[2] http://projects.kmi.open.ac.uk/akt/ref-onto/
[3] https://lov.linkeddata.es/dataset/lov/vocabs/crsw

ing the data with the schema, a connected ontology graph allows for the newly discovered shorter path (blue path) describing a Course that is taught at an Organization that, in this case, is a University. Finally, akin to the UMLS Semantic Types [14], assigning defined types to concepts from different ontologies reduces the semantic complexity of the connected KG schema by grouping related concepts. In Figure 2, the semantic type is exemplified by the red box that groups all organisation-related terms under the Organisation semantic type. This grouping opens the possibility for type-based knowledge inference and provides an additional layer of structuring for complex schemas with several ontologies connected in a single graph.

In this paper, we propose and evaluate a method that exploits existing ontologies to improve integration of data at the schema level in KGs. For this purpose, we analyse the impact of axioms and ontology matching in the overall structure of the graph. We also demonstrate how to take advantage of an ontology graph as the underlying schema of a KG by proposing a method that assigns semantically relevant groups to ontology entities.

We start the article by discussing related work, followed by a detailed explanation of the methods and evaluation. We then present and analyse the results. Finally, we present our conclusions, limitations of the methods and future work.

## 2   Related Word

DBpedia [12] is a KG built from information in Wikipedia, with an underlying ontology semi-automatically extracted from the most commonly used Wikipedia information boxes. YAGO [20] is another KG that extracts knowledge from Wikipedia, but its ontology combines the Wikipedia category system with the WordNet [15] taxonomy. The relations in YAGO are manually evaluated and some also provide links to DBPedia concepts. However, both DBPedia and YAGO restrict their knowledge to information present in Wikipedia and, therefore, their ontologies focus on describing information available on this website. Instead, we propose methods to extract a schema for a KG from multiple available ontologies, facilitating the integration of knowledge from different domains.

The problem of knowledge integration addresses interoperability problems in data. KARMA [10] is a system that models a variety of data sources, in different formats, with ontologies provided by the users. This system, however, does not explore the potential of the ontologies as an additional layer to enrich relationships. In this paper, we propose to not only use the ontologies to model the data but, also, to provide an additional layer of knowledge in a KG that can be exploited to infer new knowledge in the data layer.

KnowLife [6] is a biomedical knowledge base that integrates unstructured data. It uses UMLS Semantic Types [14] to detect entities in text and also extract relationships between these entities from the semantic types. The UMLS Metathesaurus [4] is used as a graph database related through UMLS Semantic types. In this work, we propose to take advantage of the UMLS Metathesaurus and Semantic Types not as a base for our ontology graph but as a method to enrich the knowledge in the KG.

## 3   Methods

The following sections describe our methods for building, enriching, and characterising an ontology graph to be used as the foundation of a KG. Figure 3 shows

the stages of the proposed methodology based on the hypothetical example from Figure 2. The following sections describe these stages in detail.

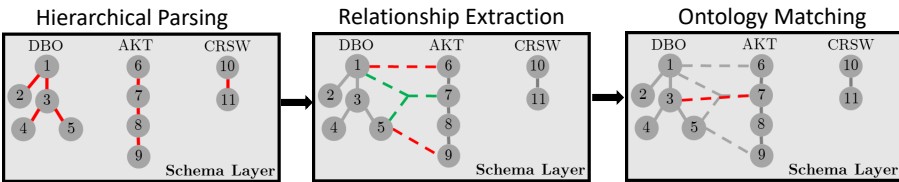

Fig. 3: Methodology stages.

## 3.1   Hierarchy Parsing

A graph $G$ is defined as $G = (V, E)$, where $V$ is a set of vertices or nodes and $E$ is a set of edges that connect the nodes.

In this step, a set of ontologies is loaded and processed into a graph $G$. The ontology classes are represented as nodes in V, while $E$ translates into the set of hierarchical relationships obtained from the *rdfs:subClassOf* axiom. In Figure 3 the red lines represent these axioms in the three example ontologies. The graph resulting from this step is named *Base Graph* or BGraph for short.

## 3.2   Relationship Extraction

When present, equivalent class axioms, property domain/range definitions, and logical definitions are added to the ontology graph. Equivalent class axioms (*owl:equivalentClass*) are added as bi-directional edges between the equivalent nodes. The property domain/range definitions are added as an edge between the domain and the range, and the edge is labelled with the property name.

Logical definitions [16] are complex axioms that refer to the relationship between a defining class, $x$, and a general class $g$ that is discriminated from other classes of $x$ with a class $d$. For example, the Cell Ontology [1] contains a logical definition for the class *cardiac neuron* (CL:0010022), where $g$ is a *neuron* (CL:0000540) that is *part of* $d$, the *heart* (UBERON:0000948). Logical definitions are translated into two edges: $E_1 = (x, g)$ and $E_2 = (x, d)$.

Figure 3 illustrates *owl:equivalentClass* axioms, in red, connecting DBO's *Organization* with AKT's equivalent class. Similarly the axioms connect *University* between the two ontologies. The diagram also shows an hypothetical logical definition (green line) to connect the intersection between DBO's *Organization* and *University* with AKT's *Higher Education Organization*. The graph resulting from this step is an increment over the *Base Graph* and was named *Axiom Graph* or AGraph for short.

## 3.3   Ontology Matching

Ontology matching is the process of finding mappings between overlapping concepts in ontologies to create alignments. We apply this approach as a edge enrichment step, however, these edges do not necessarily adhere to ontology formalisms. Instead, they represent links between related concepts to be leveraged when extracting knowledge from a KG.

AgreementMakerLight (AML) [8] is an ontology matching system that produces alignments between pairs of ontologies. This system is consistently one of

the best performing in the Ontology Alignment Evaluation Initiative (OAEI) [7], which evaluates the performance of ontology matching systems in specific tracks that benchmark their efficacy and efficiency.

We match sets of ontologies with AML's Word Matcher, which uses a bag-of-words strategy to find word overlap between two ontology class labels and scores the mappings with Jaccard similarity. Mappings with a score higher than 0.4 were selected. If conflicting mappings were found, only the highest scoring ones were selected for the final alignment. We performed a baseline evaluation of the chosen ontology matching strategy by using the anatomy dataset from the Anatomy track of the OAEI 2018[4], which provides two anatomy ontologies and a manually curated reference alignment.

Figure 3 illustrates a possible mapping (red line) between DBO's *Educational Institution* with AKT's *Educational Organization*. The graph resulting from this step is an increment over the *Axiom Graph* and was named *Mappings Graph* or MGraph for short.

### 3.4  Characterisation and Evaluation

To the best of our knowledge, no benchmark or comparable approach exists to evaluate our methods. However, the graphs obtained from the hierarchy parsing, axiom extraction, and ontology matching (BGraph, AGraph, and MGraph, respectively) are increments of each other, i.e. the AGraph is built from the BGraph and the MGraph is enriched from the AGraph. This incremental building process allows us to assess how the more complex ontology axioms and the ontology matching process affect the base structure of the ontology graph.

We adopt the BGraph as the baseline for the evaluation since it represents the minimum set of existing edges between the considered ontologies. We then evaluate each step as an increment over the previous graph with characterisation measures that provide an overview of the structure. This evaluation strategy assesses the impact of each step, giving a better understanding of the graph evolution. Ideally, the results of this evaluation present graphs that are increasingly more cohesive and connected, translating into graphs that can relate concepts more efficiently and effectively. The metrics used are defined as follows:

- **Connected Components (CC)** refers to sub-graphs where every node has a path to all other nodes in the same sub-graph.
- **Clustering Coefficient (CCF)** [22] measures the degree at which the nodes in the graph cluster together, based on triads.
- **Average Node Degree (AD)** is the average number of edges that are connected to a node.
- **Cohesiveness (CV)** [23] measures how difficult it is to split the graph.
- **Distribution of Shortest paths** measures the frequency of the lengths of the paths between two nodes in the graph such that the number of edges in these paths are minimised.

More specifically, the number of CC and the AD are directly affected with the addition/removal of edges. In the ontology graph, these metrics evaluate the overall connectedness of the graph. Unchanging values between assembly steps mean that the number of edges added had a low impact on the graph structure.

---

[4] http://oaei.ontologymatching.org/2018/anatomy

CCF and CV have a range of [0,1] and measure how closely connected are the nodes in the graph and how robust are those connections. Ideally, each step of the building process should increase the value of these measures, therefore producing an increasingly connected graph where related concepts are more clustered and harder to separate into disconnected or isolated structures.

The distribution of shortest paths evaluates the impact of the edges added in each step in terms of efficiency of the graph when querying for similar or related concepts. An increase in the number of shortest paths demonstrates that new edges are creating new paths between the concepts in the graph.

### 3.5   Semantic Type Assignment

Besides providing a standardised structured schema for a KG, is possible to extract knowledge from the ontology graph and later exploit it to enhance the KG. For this purpose we propose a method that assigns a semantic type to ontology classes. The semantic type allows ontologies to be grouped beyond their original domain into clusters of related concepts. These groups can be exploited, for example, in inference tasks for consistency checks or for information summarisation and extraction.

In this step, we associate nodes with a term obtained from specific background knowledge. For the purpose we used WordNet synonyms for the type assignment of general domain ontologies and UMLS Semantic Network [13] types for the domain-specific biomedical test case.

**WordNet semantic type assignment.** WordNet is a database of lexical elements that are grouped into sets of cognitive synonyms (synsets). In February 2019, WordNet had 117 659 unique synsets.

We assigned WordNet synsets to general domain concepts by selecting the first synset that matches the label of each ontology class. When no match is found, the label is split into its words and the first synset matching each word is added as a semantic type. Our approach was evaluated by removing manually assigned semantic tags in the SemCor corpus [11], automatically assigning a tag using the method previously described and, finally, computing precision and recall between the new and the original tags.

**UMLS semantic type assignment.** The UMLS Semantic Network consists on a set of broad concepts that categorise all concepts in the UMLS Metathesaurus. In November 2018, the UMLS Semantic Network had 127 unique semantic types.

We assigned semantic types using two methods: direct assignment and automatic extraction. When a class in an ontology cross-references a UMLS source term, we directly assign the node in the schema with the same type as the cross-referenced UMLS source term. However, when no semantic type is directly extracted, it is automatically computed as follows:

1. The preferred label, $L$, is stemmed and stop words are removed.
2. $L$ is matched with terms from the UMLS Metathesaurus by first finding all UMLS terms that match at least one word of $L$ and then selecting the terms with the highest number of intersecting words, if no perfect match is found.
3. If more than one match is found, they are scored with a cosine similarity function, and the ones with the highest scores are selected.

The approach is evaluated by removing the labels from ontology classes cross-referenced with UMLS types and re-computing them with the described approach, evaluating the original types against the new with precision and recall.

## 4 Results and Discussion

This section details and discusses the results of the assembly, enrichment and enhancement of the ontology graph.

### 4.1 Ontology loading

We performed experiments over ontologies of diverse domains found in the Linked Open Vocabularies (LOV) [21] portal. Ontologies featured in this portal are assigned subject tags. The three most common tags are "Methods", "Metadata", and "Geography". From the 659 ontologies available in the LOV in February 2019, we were able to download and parse 340. We refer to the graphs obtained from this set of ontologies as General Ontology Graph (GOG).

We also used a domain specific ontology set extracted from the Ontology Lookup Service (OLS) [9], a repository of biomedical ontologies. Out of 220 ontologies available through the OLS REST API on February 2019, we were able to download and parse 206. The most common reason for discarding an ontology was the presence of outdated information, e.g., *owl:import* statements referring to ontologies that do not exist anymore. We refer to the graphs obtained from this set of ontologies as Biomedical Ontology Graph (BOG).

These two test sets provide a rich environment for testing with ontologies with different characteristics. Biomedical ontologies are domain-specific ontologies and, therefore, are more likely to have equivalent or related terms between them. These ontologies are also characterised by their well-defined standards [19], large size, and complex axioms. Ontologies found in the LOV have diverse domains and, therefore, the alignment of the topics is not guaranteed. These ontologies have different levels of formalism and mostly follow LOD guidelines.

### 4.2 Ontology Matching

The baseline evaluation ontology matching strategy obtained a precision of 53.6% and a recall of 83.2% when matching the two ground-truth anatomy ontologies. Despite the reasonably good performance with this matching strategy, a precision of 53.6% can lead to a significant number of incorrect relations in the graph. However, the OAEI Anatomy track evaluates the ability of an ontology matching system to find complete equivalences between two ontologies. In our case, we are looking for related concepts that do not need to be equivalent, e.g., the matching approach aligns *lateral thyrohyoid ligament* with *thyrohyoid*. These two concepts are not equivalent and, therefore, the OAEI Anatomy reference alignment does not contain this mapping. However, these two concepts are related since *lateral thyrohyoid ligament* is conceptually related to *thyrohyoid*. Therefore, in the case of building a KG, it is still desirable to find this relationship since it can facilitate the connection between two different concepts that are not semantically equivalent but have a real-world anatomical relationship.

Table 1 shows the results of the pairwise ontology matching over the general and the biomedical sets of ontologies. The general set of ontologies obtained more alignments than the biomedical set. However, this difference is due to the difference in the number of ontologies in each set since the median number

| $O$ | $|O|$ | $|A|$ | $med(|A|)$ | $avg(sim)$ |
|---|---|---|---|---|
| General ontologies | 420 | 33 619 | 2 | 0.66 |
| Biomedical ontologies | 220 | 14 292 | 15 | 0.53 |

Table 1: Results of the ontology matching process. $O$ is the set of ontologies, $A$ is the resulting alignment, $med(A)$ is the median number of mappings in an alignment, and $avg(sim)$ is the average Jaccard similarity of the mappings.

of mappings in the general set was significantly lower than the biomedical set. These results show that, as expected, ontologies in a more restricted domain will find a more significant number of overlapping or related concepts. The average Jaccard similarity in the general set of ontologies was 66%, while the biomedical set obtained 53%.

### 4.3   Characterisation and Evaluation

Table 2 compares the ontology graphs over different stages of construction. In both GOG and BOG, the number of nodes shows a small increase from the BGraph to the AGraph due to axioms that reference ontology classes outside of the scope of the ontology set. In these cases, the new ontology classes are added as new nodes to the AGraph.

The GOG shows an increase of ≈21% in the number of edges between the BGraph and the AGraph and ≈56% more edges in the MGraph than in the AGraph. In the BGraph, ≈48% of the nodes are in the Largest Connected Component (LCC), but this number increases to ≈62% in the AGraph and ≈88% in the MGraph. Most of the elements not connected to the LCC are isolated, forming a new connected component with a single node and no incoming or outgoing edges. The most common reason for disconnected nodes in the GOG are inconsistencies in the definition of classes, properties and their relations. For example, the class `http://purl.obolibrary.org/obo/HAO_0002311` is a root class with no descendants and, therefore, is isolated in the graph.

Overall, in the BOG, the BGraph and the AGraph are structurally similar since only a small number of edges was added (≈7% more edges). However, between the AGraph and the MGraph, the differences are more prominent due to an ≈48.9% increase in the number of edges. The BGraph and AGraph have ≈94%

| Graph | Stage | $|\mathbf{V}|$ | $|\mathbf{E}|$ | LCC | $|\mathbf{CC}|$ | $|\mathbf{1\text{-}CC}|$ | CCF | AD | CV |
|---|---|---|---|---|---|---|---|---|---|
| GOG | BGraph | 16 510 | 16 755 | 7 889 | 3 210 | 2 867 | 0.022 | 2.450 | 0.433 |
| | AGraph | 16 580 | 21 141 | 10 307 | 2 252 | 2 071 | 0.065 | 2.757 | 0.576 |
| | MGraph | 16 580 | 47 727 | 14 634 | 1 517 | 1 553 | 0.177 | 5.639 | 0.886 |
| BOG | BGraph | 4 502 980 | 6 967 165 | 4 223 817 | 66 270 | 65 392 | 0.049 | 3.089 | 0.562 |
| | AGraph | 4 502 981 | 7 456 127 | 4 227 375 | 63 522 | 62 693 | 0.050 | 3.160 | 0.575 |
| | MGraph | 4 502 981 | 14 595 189 | 4 458 547 | 41 317 | 40 581 | 0.129 | 5.978 | 0.833 |

Table 2: Characterisation of the GOG and BOG. $V$ is the set of nodes and $E$ is the set of edges. LCC - Largest Connected Component; CC - Connected Components; 1-CC - single node CC; CCF - Clustering Coefficient; AD - Average Node Degree; CV - Cohesiveness

of the nodes in the LCC. In the MGraph, $\approx 99\%$ of the nodes are connected in the LCC. Most of the single elements connected components are due to ontologies that do not follow common standards and formalisms to build an ontology or reuse classes. For example, the ontology *Flora Phenotype Ontology*[5] was created only with classes and contains no intra or interlinks between classes.

Figure 4a and Figure 4b show the distribution of shortest paths for the ontology graphs. Due to the large size of the BOG, the distance histogram was computed for a sample of 60% of the total number of nodes of this graph.

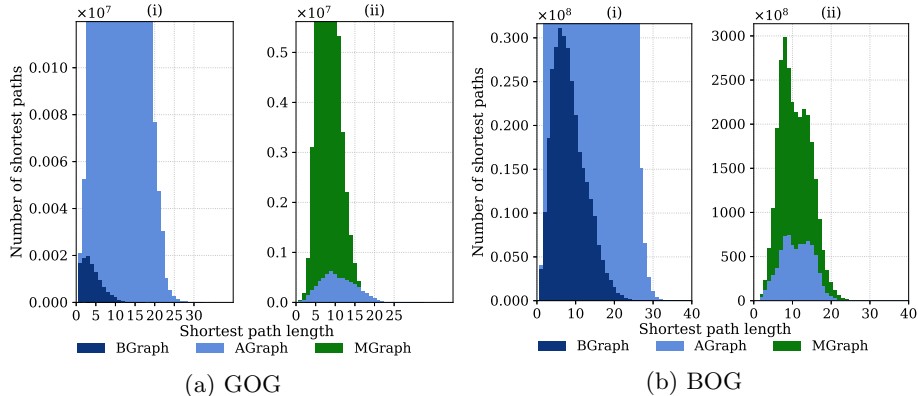

Fig. 4: Distribution of shortest paths.

In both figures, plot (i) corresponds to the overlap of the distance histogram of the BGraph with the AGraph, and plot (ii) corresponds to the overlap of AGraph and MGraph. Overall, both axiom and mapping edges have a significant impact in the number of shortest paths that connect the nodes, which is directly related with the increase of the number of elements connected in the LCC, i.e., more nodes connected, new shortest paths created. Despite the lower impact of the edges added to AGraph in terms of structural properties in comparison to MGraph, the distribution of shortest paths shows a comparable increase in the number of shortest paths.

These new paths facilitate the discovery of relationships between concepts in the graph, but also help shorten distances between them. More concretely, the changes in the distribution of shortest paths can be illustrated, for example, by the changes in the shortest path between the node with the label *cancer* (DOID:162) and *malignant cell* (CL:0001064). In BGraph, the shortest path between the two nodes is 10, in AGraph is 4, and in MGraph the concepts nodes are directly connected, i.e. a shortest path length of 1.

In summary, the characterisation of the proposed ontology graph demonstrates the contribution of each of the building steps to the overall structure of the graph. The addition of axioms and mappings improves the connectedness of the graph and creates new paths between nodes. The ontology matching step proved to be significant, which shows that, in the context of building a foundation for a KG, ontology matching is an approach that can connect previously disconnected parts of the graph facilitating the process of finding related concepts. The new edges allow concepts in the graph to be more easily reachable between each other, facilitating the discovery of new relationships.

---

[5] `http://purl.obolibrary.org/obo/flopo.owl`

### 4.4  Semantic types

The evaluation of the methodology for WordNet type assignment achieved a precision of 63.2% and a recall of 77.1%. Considering the reasonable performance of the naive approach, we proceeded to the assignment of types to the ontology classes in the GOG. The approach identified at least one type for 73.6% of the nodes. The most common types were *item.n.01* with 1 562 nodes, *assay.n.01* with 120, and *biological.a.01* with 99 nodes. The most common types are broad terms due to the assignment of all synsets that match each word of the label to the ontology class. A manual inspection revealed that this implementation detail was also responsible for most of the wrong type assignments.

For example, the synset *person.n.01* is defined in WordNet as "a human being". The automatic matcher assigned 29 ontology classes to this synset. A manual analysis showed that 75.8% of the classes assigned to this synset closely matched the WordNet concept of person or were related to it. For example, `http://xmlns.com/foaf/0.1/Person`, `http://d-nb.info/standards/elementset/gnd#DifferentiatedPerson`, and `http://purl.org/goodrelations/v1#Individual` are three classes that are grouped under the synset person. Five of the explicitly seven wrong assignments occur due to the methodology of matching each word separately if no synset is found. For example, *Individual epitope immunization in vivo* (`http://purl.obolibrary.org/obo/OBI_0001176`) matched due to the keyword "individual".

The UMLS evaluation obtained a precision of 85.1% and a recall of 99%. Both strategies obtained a reasonably high performance, with the approach developed for UMLS performing better than the WordNet method. In relation to the semantic type assignment in the biomedical ontology graph, ≈3.3% nodes had a direct semantic type assigned. The automatic approach was able to find a type match for ≈76.9% of the remaining nodes. The most frequent types were *Eukaryote* (T204) with 724 749 nodes, *Bacterium* (T007) with 525 529 nodes, and *Plant* (T002) with 247 825.

Overall, the goal of the semantic type assignment was to provide an example of an approach to extract information from the ontology graph as a KG schema. In this context, this semantic grouping system clusters concepts into topics and, considering that both WordNet and UMLS have their own graph structures, these groups can be further exploited to extract more relationships. For example, the semantic type *Body Part, Organ, or Organ Component* (T023) groups every relevant concept under the same topic. In a search engine scenario, this semantic type can be used to retrieve all the information of the KG about body parts.

## 5  Conclusion

This work proposes methods to create an ontology graph that can be exploited to enhance the knowledge contained in a KG. We analyse these methods by comparing the evolution of the graph at different stages and assess the impact of each incremental step in the overall structure of the graph. We add edges to the ontology graph in three steps: *owl:subClassOf* axioms, remaining class axioms, and ontology alignments. Each step yields a more tightly connected and cohesive graph than the step before. The addition of the extra class axioms over the basic *owl:subClassOf* hierarchy shows that complex axioms have a significant impact on the overall graph structure, specially the number of paths between nodes. In

the context of a KG schema, taking advantage of these axioms provides more relationships between concepts that contribute to the global usefulness of a KG.

We also propose to enrich the original ontology by performing pairwise ontology matching over all the ontologies in the graph. This approach showed to have a significant impact in the structure of the graph, especially by connecting previously disconnected portions of the graph, enabling the discovery of new relationships and paths between concepts.

Finally, we present a method to exploit the proposed schema by assigning semantic types to the ontology classes. We use background knowledge extracted from WordNet and the UMLS Semantic Network to group ontology classes into their more specific domains. This approach showcases the potential to extract information from the ontology schema that can be leveraged in different scenarios, such as finding new relationships between sets of concepts.

## 6  Limitations and Future Work

In this paper, we proposed and discussed methods that can be leveraged in different scenarios, however, these results also highlighted several limitations to the approach that will be improved in the future.

We currently only consider object properties if they have domain and range assigned. In these cases, the property is added as the label of the edge between its domain and range. In the future, we will analyse how to leverage ontology properties has a new assembly step in the graph building process.

Currently, logical definitions are added as two separate edges, when in reality these definitions represent an intersection between three concepts. Therefore, the integration of logical definitions as edges will be improved to more closely represent this relationship. In the future, we will explore how logical definitions can be fully explored, including assessing the potential of more complex ontology matching to discover new relationships involving three concepts [17].

Despite the optimisation of the ontology matching step, ontology matching is a complex subject which was simplified for the purpose of evaluating its applicability in this case. In the future, we will examine how to optimise this process further, including developing an heuristic for large scale ontology matching. A robust ontology matching step can also be exploited to make a more compact graph by merging equivalent concepts in a single node.

In relation to the semantic type assignment, currently we propose naive approaches that achieve reasonably good results. However, a fully automated and reliable process would be ideal for semantic type assignment. Further applications for querying and relationship prediction will be developed to extend the usefulness of the ontology graph schema. Overall, the approaches proposed and the analysis show the potential of a unified ontology graph as KG schema to knowledge extraction and relationship discovery.

## Acknowledgements

This work has been partly funded by Science Foundation Ireland (SFI) under Grant Number SFI/12/RC/2289, Insight Centre for Data Analytics.

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
