# OpenReview forum: "Leveraging Ontologies for Knowledge Graph Schemas"
_eswc-conferences.org/ESWC/2019/Workshop/KGB — KGB 2019_

### Official Review · ~Craig_Knoblock1 · 2019-03-28
**Not convinced by the problem being solved in this paper**

**Rating:** 2
**Confidence:** 3

**Review:**

This paper presents an approach to combining multiple ontologies to build the schema for a knowledge graph.  While the paper is well written, I'm a bit skeptical of the problem being solved in this paper.  The motivating example combines an ontology about a university with an ontology of courses by finding the overlap in these two ontologies and then deriving shorter paths between some of the ontology elements.  This example is not at all compelling given that any ontologies that I seen for universities already include courses.  In my experience, it is sometimes necessary to combine ontologies to build a particular knowledge graph, but overlap in the ontologies is usually a problem since the knowledge graph builder then gets confused about which concepts to use.  So in our work we would only combine complimentary ontologies and linking them together would not usually be necessary since they would cover different parts of a domain.

The techniques presented in this paper look for overlap in the classes and try to derive additional links between the two ontologies.  Certainly finding the overlap is important since you don't want to have two classes for the same concept in the final schema.  But I'm not convinced that deriving additional links is a good idea and there is no evaluation in the paper that shows that this is useful.

The section on semantic typing in the paper was not well-motivated and it did not seem to fit well in the rest of the paper.

The evaluation looked at pairwise combinations of a wide variety of ontologies and then evaluated how well the methods produced a combined ontologies with some specific properties described in the paper.  But there was no evaluation of these specific properties that would justify why they are the right properties to optimize to build a schema for a knowledge graph.

I would suggest that the authors work a real-world examples where the combination of several ontologies would actually be needed to create a knowledge graph schema.  Then take that example and use it throughout the paper to explain your method and show that you method would combine the two ontologies in a meaningful way and derive the properties that you would actually want in the final knowledge graph.

---

### Official Review · ~Anastasia_Dimou1 · 2019-04-02
**a starting point for a hard-to-tackle problem**

**Rating:** 3
**Confidence:** 2

**Review:**

This paper is about a method to build a schema for knowledge graphs that integrates ontologies in a single unified graph. The method consist of the stage of assembling the ontologies (which on their own turn consists of hierarchy parsing and relationship extraction) and the stages of edge refinement, relying on ontology matching, and semantic type attribution, based on domain-specific services, e.g. UMLS and WordNet.

The paper is well written and tries to solve a hard-to-tackle problem, let alone evaluate. I am not very confident about three aspects: 1. I am not sure if ontology matching techniques are ideal for aligning complementary ontologies, but mostly 2. I am not sure if the extended graph is validated in the end, namely does the edge refinement introduce inconsistencies? and 3. I am wondering if it is safe to consider the base graph as the baseline for the evaluation.

However, it is a workshop paper and, given that we rarely encounter such solutions, I would suggest to accept the paper and hopefully trigger relevant discussions!

---

### Decision · Program_Chairs · 2019-04-08
**Acceptance Decision**

**Decision:**

Accept

**Comment:**

This contribution is accepted for presentation at the KGB2019 workshop, and for inclusion in its proceedings.